# Optimization of the Winding Layer Structure of High-Pressure Composite Overwrapped Pressure Vessels

**DOI:** 10.3390/ma16072713

**Published:** 2023-03-29

**Authors:** Chengrui Di, Bo Zhu, Xiangji Guo, Junwei Yu, Yanbin Zhao, Kun Qiao

**Affiliations:** 1School of Materials Science and Engineering, Shandong University, Ji’nan 250061, China; 2Shandong Special Equipment Inspection and Testing Group Co., Ltd., Ji’nan 250101, China; 3Shandong Huate Tianwei New Material Co., Ltd., Dezhou 251114, China; 4School of Mechanical, Electrical & Information Engineering, Shandong University, Weihai 264209, China

**Keywords:** winding, COPV, fiber orientations, finite element

## Abstract

The large thickness COPV is designed by netting theory and the finite element simulation method, but the actual performance is low and the cylinder performance still cannot be improved after increasing the thickness of the composite winding layer. This paper analyzes the reasons for this and puts forward a feasible solution: without changing the thickness of the winding layer, the performance of COPV can be effectively increased by increasing the proportion of annular winding fiber. This method has been verified by tests and is supported by theory.

## 1. Introduction

A composite overwrapped pressure vessel (COPV) has the advantages of high strength–weight ratio, excellent fatigue resistance, leakage characteristics before explosion, and flexibility in structural design [1,2]. In recent years, COPVs have been widely used in aerospace, new energy vehicles, and chemical and mining industries.

Typically, a COPV consists of two parts: a metal or plastic liner that determines the shape of the COPV and a composite winding layer on the outer surface of the liner. The liner mainly plays a role in maintaining the shape of the COPV and sealing the gas, while the composite winding layer is the key structure to determine the bearing capacity of COPV [3,4]. At present, COPV design theories mainly include laminate theory, netting theory, membrane structure theory, etc. [5,6], among which the netting theory is the most used. In addition, using finite element simulation software such as ANSYS and ABAQUS is a necessary method to design and optimize the COPV [7,8]. Among the many features of COPVs, bearing capacity and fatigue performance are major concerns for COPV developers and users [9,10].

According to the bearing capacity division, COPV mainly has 35 MPa and 70 MPa specifications; 70 MPa COPV has a higher bearing capacity and can store more energy, so it is the focus of researchers’ attention.

To further improve COPV performance, based on these theories, researchers carried out studies on the effects of fiber strength [7], fiber direction (winding mode, winding angle, etc.) [8], resin properties (mechanical, thermal, and process properties, etc.) [11], winding process (tension, speed, yarn number, etc.) [12,13], and other factors [14,15,16,17,18] on COPV properties. 

According to the netting theory, under the condition of vessel size invariability, increasing the bearing capacity is mainly achieved by increasing COPV winding layer thickness. This method is effective for thin-walled structures but has limitations for the design and calculation of large-thickness 70 MPa COPVs, and simply increasing the thickness of the winding layer will increase the weight of COPV, resulting in an increase in cost [19]. Moreover, in many studies [20,21,22] on COPVs, theoretical formulas and finite element software are used to design and predict performance, and then the COPV performance is measured to verify the accuracy of theoretical design and simulation. Some studies discuss the difference between theoretical design and practical performance and give effective methods to combine design and practical experience.

In this work, by comparing the theoretical calculation, finite element simulation results, and actual test performance of COPVs, based on the failure mode of COPV, a method to optimize the performance of large-thickness COPVs is proposed which provides design ideas and experience parameters for the design and manufacture of 70 MPa COPVs.

## 2. Materials and Methods

### 2.1. Liner Structure and Material Performance

The liner is made of 6061 aluminum alloy. The size of the liner and the fiber winding direction are shown in Figure 1. The properties of the materials used are shown in Table 1. The winding resin uses the winding resin system reported in the paper [11].

### 2.2. Thickness of Composite Layer

#### 2.2.1. COPV Cylindrical Section

The winding layer structure design of the COPV’s cylindrical section is calculated by netting theory, which is based on the following assumptions [5]: (1) in COPVs, only the fibers in the wound composite layer bear the load, and the fibers only bear the load along the fiber direction; (2) each fiber has the same strength; (3) the influence of the matrix resin is not considered; (4) the influence of winding tension and uneven fiber stress is not considered.

The basic formulas are shown in (1)–(5):(1)$${\sin\alpha }_{c}=\frac{r_{0}}{R}$$
(2)$$h_{A}=\frac{P_{b}\cdot R}{{2K\sigma \cos}^{2}\alpha_{c}}$$
(3)$$h_{H}=\frac{P_{b}\cdot R}{2K\sigma }\left({2-\mathrm{tg}}^{2}\alpha_{c}\right)$$
(4)$$N_{a}=\frac{h_{A}}{m}$$
(5)$$N_{h}=\frac{h_{H}}{m}$$

Symbols are as follows: $\alpha_{c}$: the winding angle; *r*_0_: the radius of the COPV polar hole in mm; R: the radius of the cylindrical part in mm; $P_{b}$: the burst pressure in MPa;$\;h_{A}$: the thickness of the annular layers in mm; $h_{H}$: the thickness of the helical layers in mm; σ: the fiber strength in MPa; *K*: the fiber strength utilization rate—0.5 ≤ *K* < 1; *N_a_*: number of annular fiber layers; *N_h_*: number of helical fiber layers.

The values of *N_a_* and *N_h_* are taken as the minimum even number greater than the calculation result. For example, if the number of annular fiber winding layers calculated in this paper is 24.4, then *N_a_* is taken as 26, and the ratio of the annular fiber layer thickness to the helical fiber layer thickness is written as *λ*, so *λ* = *N_a_/N_h_*. According to the actual measurement, the single layer thickness (*m*) of the annular and helical winding fibers (dry yarn) during winding is 0.18 mm.

#### 2.2.2. COPV Dome Section

The fiber thickness and winding angle of the winding layer in the dome section of the finite element analysis model in this paper are calculated by Formulas (6) and (7). The structure of the dome section is shown in Figure 2.
(6)$$\alpha_{f}=\sin^{-1}\frac{r_{0}}{r}$$
(7)$$h_{f}=h_{A}\sqrt{\frac{R^{2}-r_{0}^{2}}{r^{2}-r_{0}^{2}}}$$

Symbols are as follows:$\;\alpha_{f}:$ the winding angle of dome section;$\;h_{f}\;$: the thickness of fiber layer at the dome; *h_A_*: the thickness of the fiber layer in the cylindrical section; *r*_0_: the radius of the polar hole; *R*: the outer radius of the cylindrical section of the liner; *r*: the radius of the concentric circle of the dome.

### 2.3. Design Pressure

The rated working pressure of the COPV designed in this paper is 70 MPa. According to GB/T 35544-2017, the minimum burst pressure of the COPV shall be 2.25 times of the rated working pressure, and the design pressure of the 70 MPa COPV shall not be less than 157.5 MPa. Therefore, the design burst pressure of the COPV in this paper is 160 MPa.

### 2.4. Manufacturing Method

In this paper, 70 MPa COPVs were prepared by wet winding process. During the winding, the annular fibers and the helical fibers are alternately wound and the winding of helical fibers is carried out by using approximate geodesics. It can be seen from Formula (5) that in the dome section the thickness $(h_{f}$) of the fiber winding layer changes with the concentric circle radius (*r*) of the dome, and the change trend is shown in Figure 3. The closer to the polar hole, the greater the thickness of the fiber layer. In the actual winding process, in order to prevent the fibers near the polar hole from accumulating and becoming too thick, the designers usually use the reaming process for winding helical fibers; that is, the winding angle changes with the number of winding layers so that the fibers near the pole hole are evenly distributed.

According to Formula (3), it can be calculated that the winding angle is 10.4°, and the fiber winding angle $(\alpha_{c}$) in the reaming process used in this paper varies between 10.5° and 15°. According to Formulas (1)–(3), when the winding angle $\alpha_{c}$ is 10.5–15°, the variation range of the annular fiber thickness $\left(h_{A}\right)$ and the helical fiber thickness ${(h}_{H})$ of the cylindrical section of the COPV is 3.53 *K* mm–3.46 *K* mm and 1.86 *K* mm–1.92 *K* mm, respectively, where *K* is the fiber strength utilization rate; in general, 0.5 ≤ *K* < 1. From this, we can conclude that when the approximate geodesic winding is used, the small change in the winding angle $\left(\alpha_{c}\right)$ has little effect on the change in the fiber thickness in each direction.

### 2.5. Finite Element Analysis and Performance Testing

In this paper, the finite element software ANSYS is used to analyze the stress of the COPV under the internal pressure of 160 MPa. Since the COPV is an axisymmetric structure, the 1/8 part of the COPV is selected as the research object. In the finite element model, SOLID95 element and SHELL181 element are used to analyze the liner and wound layer, respectively. Symmetry constraint is applied on the profile of the cylinder model, and axial displacement constraint and rotation constraint of the other two axes are applied to the mouth of the cylinder. The hydraulic blasting test and fatigue test were carried out according to the Chinese standard GB/T 35544-2017.

## 3. Results and Discussion

### 3.1. Effect of Winding Thickness on Burst Pressure

It can be seen from Formulas (1)–(5) that when the design pressure, liner size, carbon fiber performance, and other parameters are determined, the calculated annular and helical fiber layer thicknesses (*h_A_* and *h_H_*) are only functions of the fiber strength utilization rate (*K*). The thickness of the wound fiber layer can be adjusted by adjusting the value of the fiber strength utilization rate (*K*).

The first principal stress nephograms of the winding layers in COPVs 1# and 2# are shown in Figure 4, and the COPVs after blasting are shown in Figure 5.

It can be seen from Table 2 and Figure 4 that when the COPV is subjected to an internal pressure of 160 MPa, the maximum stress of COPVs 1# and 2# is located at the cylindrical section, with the maximum values of 2189.46 MPa and 1977.26 MPa, respectively, both of which were lower than the tensile strength (2300 MPa) of T700SC/epoxy wound layer. It shows that increasing the thickness of the winding layer can reduce the stress of the COPV, thus improving the bearing capacity of the COPV. After the hydraulic burst test, the actual burst pressure of COPV 1# is 138 MPa, while COPV 2# is only 143 MPa. Both of the failure positions are in the cylindrical section of the COPVs, as shown in Figure 5, which is consistent with the maximum stress position of the finite element analysis.

From the test results of COPVs 1# and 2#, there is a large deviation between the theoretical design goal and the actual results. Although the finite element simulation can predict the maximum stress position, there are still many uncertain factors in the accurate prediction of the burst strength of COPVs. This is because netting theory is based on many ideal assumptions and the finite element analysis simplifies the model of the COPV. Both ignore the influence of various unfavorable factors in the preparation process of COPVs. Especially for the winding composite products with large thickness and working pressure of 70 MPa, the number of defects will increase with the increase in the thickness of the composite layer, and the composite layer is more likely to fail. In the curing process, the internal stress is generated due to the inconsistency of the internal and external rising and cooling rate and curing degree of the large-thickness composite, which leads to the decrease in the strength of the composite layer. At the same time, due to the control of winding tension, it will not only affect the compactness and resin content of the winding layer, but also affect the tightness of the internal and external fibers so that the internal and external fibers cannot bear the load at the same time [14], thus reducing the burst pressure of the COPV. These factors are difficult to reflect in the netting theory and finite element simulation calculation, resulting in a gap between the theoretical calculation results and the actual results.

In addition, we can see from the hydraulic burst test results that although the number of winding layers of COPV 2# is increased by six layers compared with COPV 1#, the burst pressure of the COPV is only increased by 5 MPa, indicating that as the composite layer becomes thicker, the strength utilization ratio of the fiber decreases, and it is difficult to improve the bearing capacity of the COPV simply by increasing the thickness of the fiber layer. It can be seen from Figure 5 that both COPV failure modes are longitudinal tearing in the cylindrical section, indicating that the annular fiber strength in the cylindrical section is not enough, leading to the fracture of the annular fiber layer under high internal pressure. Therefore, if the thickness of the annular fiber layer can be appropriately increased, the bursting pressure of the COPV can be effectively increased. 

### 3.2. Effect of Ratio of Annular/Helical Fiber on Bursting Pressure of COPV

Based on the design of COPV 2#, we adjusted the fiber direction; that is, under the condition of keeping the total number of fiber layers unchanged, we adjusted the ratio (*λ*) of the number of annular and helical fiber layers of the COPV. The specific results are shown in Table 3.

The first principal stress nephograms of COPVs 3#–5# are shown in Figure 6, and the failure modes of COPVs 3#–5# are shown in Figure 7.

It can be seen from Table 3 and Figure 6 that when the total number of winding fiber layers of COPV is the same, the maximum first principal stress of the COPV is 1977.26 MPa, 1716.77 MPa, 1614.55 MPa, and 1935.90 MPa, respectively when the *λ* increases from 2.0 to 2.4, 3.0, and 3.8. The maximum stress of the COPV decreases first and then increases with the increase in the *λ*. The burst pressure of the COPV is opposite to the maximum stress, as shown in Figure 8.

This indicates that the bearing capacity of the COPV can be significantly improved by properly increasing the *λ*. When the *λ* is 3.0, the maximum stress value of the COPV is the smallest and the burst pressure is the largest.

We can also draw the corresponding conclusion from the derivation of the netting theory formula. Formula (8) can be derived from the netting theory Formulas (2) and (3):(8)$$\lambda =\frac{h_{H}}{h_{A}}=3\;\cos^{2}\alpha_{c}-1\;(0{}^{\circ}\;<\;\alpha_{c}\leq \;90{}^{\circ})$$

From the formula ${\;\sin\alpha }_{c}=\frac{r_{0}}{R}$ (0° < α*_c_* ≤ 90°) (1), we can know that α_*c*_ is an increasing function of $\frac{r_{0}}{R}$, and when $\;\frac{r_{0}}{R}$ is smaller, the winding angle *α_c_* is also smaller. According to Formula (8), *λ* is a decreasing function of *α_c_*. When the winding angle (*α_c_*) is smaller, the ratio (*λ*) is larger. According to Formulas (1) and (8), it can be concluded that when the polar hole radius (*r*_0_) (the polar hole radius in this paper is the COPV mouth radius) is constant, *λ* increases with the COPV radius (*R*); that is, the proportion of annular fiber should be larger. 

When designing COPVs with large thickness, if the wound composite layer is regarded as a combination of multiple cylinders (the cylinder radius is *R_n_*, *n* = 1, 2, 3…), *R_n_* also increases with the increase in the winding layer thickness, as shown in Figure 8. In this paper, the approximate geodesic winding with small angle is adopted, and the radius of the polar hole (*r*_0_) can be considered as a constant value. Therefore, according to the above conclusion, the ratio (*λ*) should also be larger. It can be seen that when preparing COPVs with large thickness, the ratio (*λ*) can be appropriately improved to meet the requirements of netting theory. 

However, when the ratio (*λ*) continues to increase to 3.8, the maximum stress value of the COPV increases instead, and the burst pressure decreases, indicating that when the helical fiber layer is gradually reduced, the longitudinal fiber strength of the COPV is also reduced; and when the internal pressure load of the COPV reaches a certain value, the longitudinal fiber will first fracture and fail, and the failure mode of circumferential fracture appears. For example, the failure mode of COPV 5# is that the longitudinal fiber first breaks at the shoulder with the greatest stress and then circumferential cracking occurs. Therefore, the design of COPV winding fiber layers needs to find the appropriate balance ratio of annular strength and longitudinal strength and ensure that the fiber strength in all directions can play an effective role at the same time so as to optimize the bearing capacity of the COPV. 

### 3.3. Effect of Ratio of Annular/Helical Fiber on Fatigue Property of COPVs

We also tested the fatigue of COPVs at different ratios of annular/helical fiber and investigated its influence on the location and mode of fatigue failure. The specific test results are shown in Table 4 and Figure 9.

According to the above test results, when the *λ* increases gradually, the fatigue property is also gradually improved, but when the proportion of annular fiber is too large (*λ* ≥ 3.8), the fatigue property decreases; this law is consistent with the blasting property. Further investigation revealed that the fatigue failure of the COPV was mostly due to the fracture of the cylindrical section of the liner, and the high-pressure water shot through the composite winding layer, resulting in the termination of the test. The dome section has no problems, indicating that the fatigue performance of the dome section of the liner is better than that of the cylindrical section, which is consistent with the conclusions in the literature [16].

When the ratio *λ* < 3.8, the liner failure cracks along the longitudinal expansion, indicating that when the liner is under internal pressure, the annular fiber to give the liner compressive stress is not enough, and the liner is prone to crack in the longitudinal direction. When the ratio (*λ*) is 3.8, the failure mode of the COPV is the annular crack in the cylindrical section, indicating that the longitudinal stress of the winding layer is low and the longitudinal expansion of the liner is large during the test, so the annular crack is produced.

## 4. Conclusions

According to the netting theory formula, the design parameters of the 70 MPa COPV were preliminarily determined and the failure location of the COPV was predicted by ANSYS finite element analysis method.It is found that the measured performance of the COPV is very different from the design goal, and the effect is minor when the thickness of the winding layer is increased. By analyzing the failure mode, the ratio of annular fiber to helical fiber is further adjusted, which greatly improves the bearing capacity and fatigue performance of the COPV. When the number of winding layers is 48 and the ratio of annular fiber to helical fiber is 3.0, the performance of the COPV is optimal.The method is validated by deducing the netting theory formula. However, further study is needed to determine how to use the finite element simulation method to further design and predict the performance of COPVs with large thickness.

## Figures and Tables

**Figure 1 materials-16-02713-f001:**
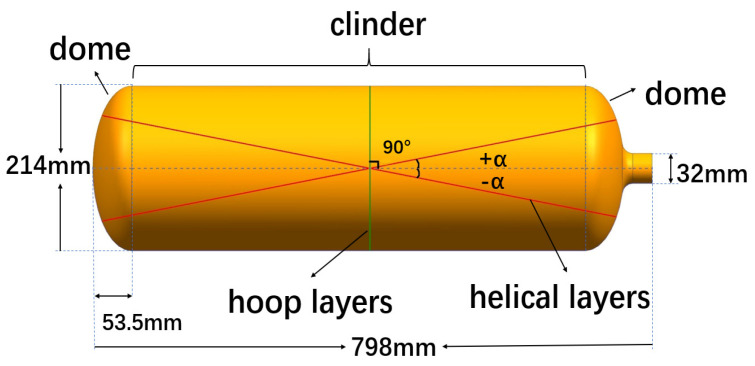
Size of liner and filament winding direction.

**Figure 2 materials-16-02713-f002:**
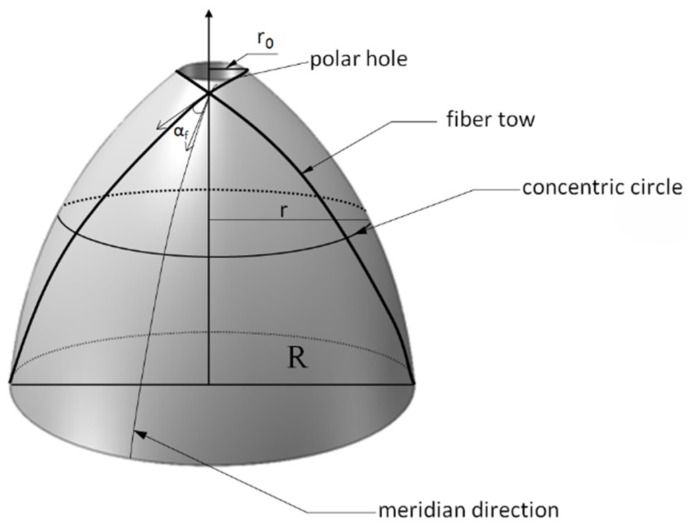
The dome section of the COPV.

**Figure 3 materials-16-02713-f003:**
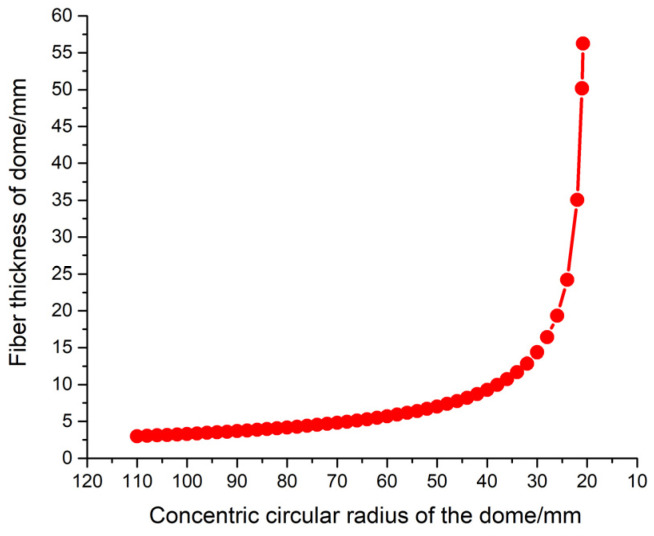
The variation trend of dome fiber thickness according to the concentric circle radius.

**Figure 4 materials-16-02713-f004:**
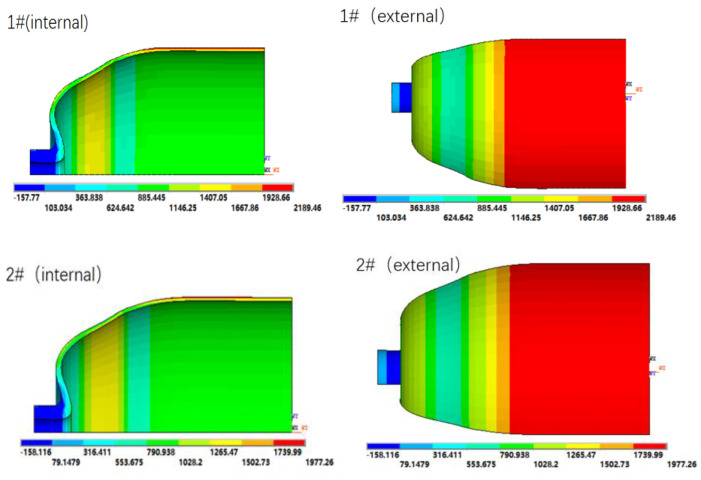
First principal stress nephograms of COPVs 1# and 2# (internal pressure: 160 MPa).

**Figure 5 materials-16-02713-f005:**
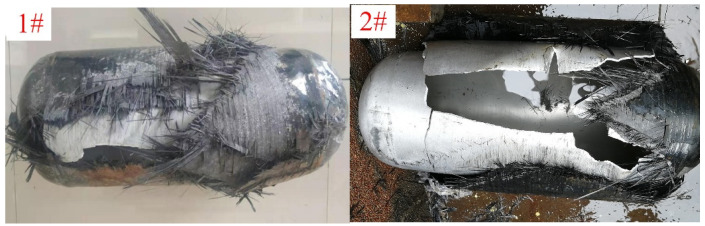
Failure modes of COPVs 1# and 2#.

**Figure 6 materials-16-02713-f006:**
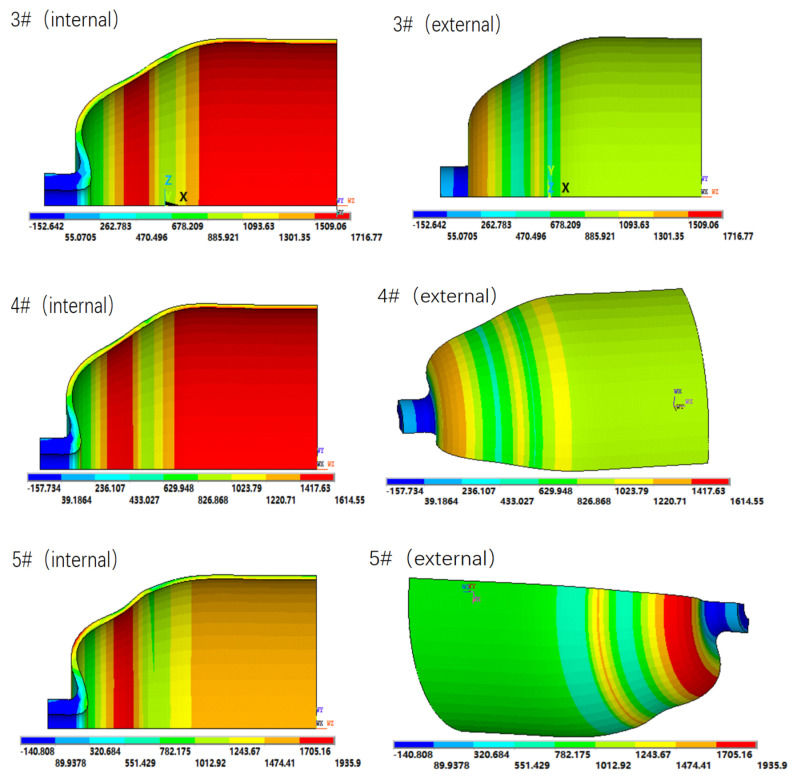
First principal stress nephograms of COPVs 3#–5# (internal pressure: 160 MPa).

**Figure 7 materials-16-02713-f007:**
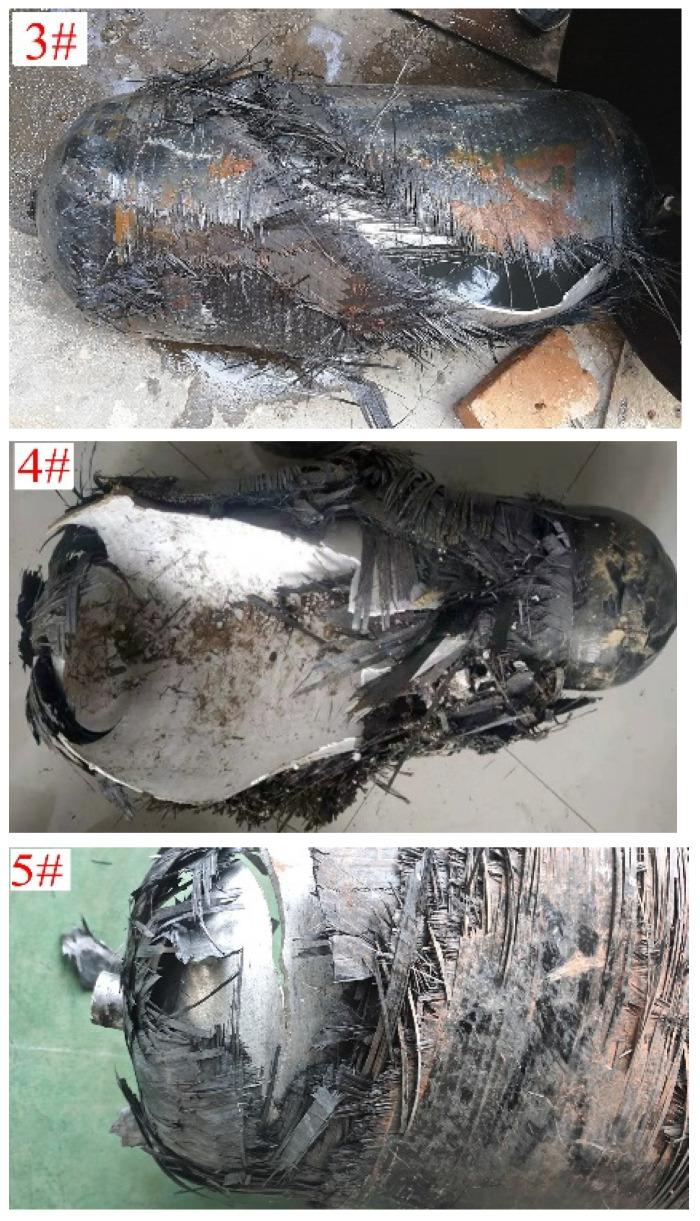
Failure modes of COPVs 3#–5#.

**Figure 8 materials-16-02713-f008:**
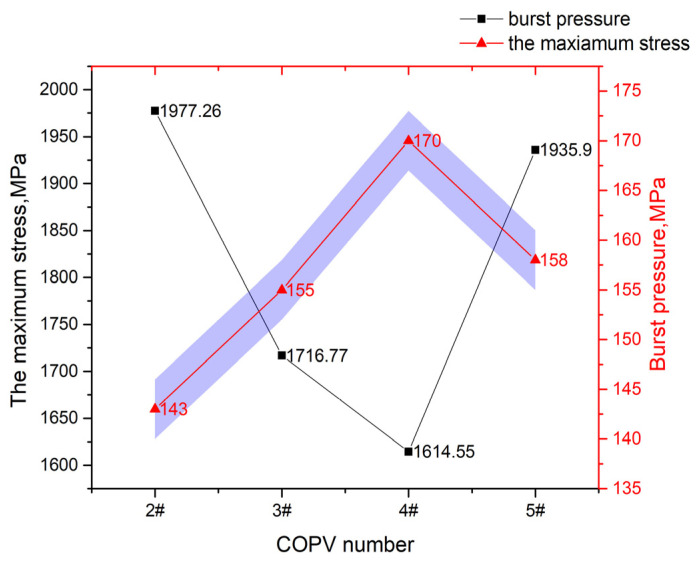
Maximum Stress Value and Burst Pressure of COPV (purple area: margin of error).

**Figure 9 materials-16-02713-f009:**
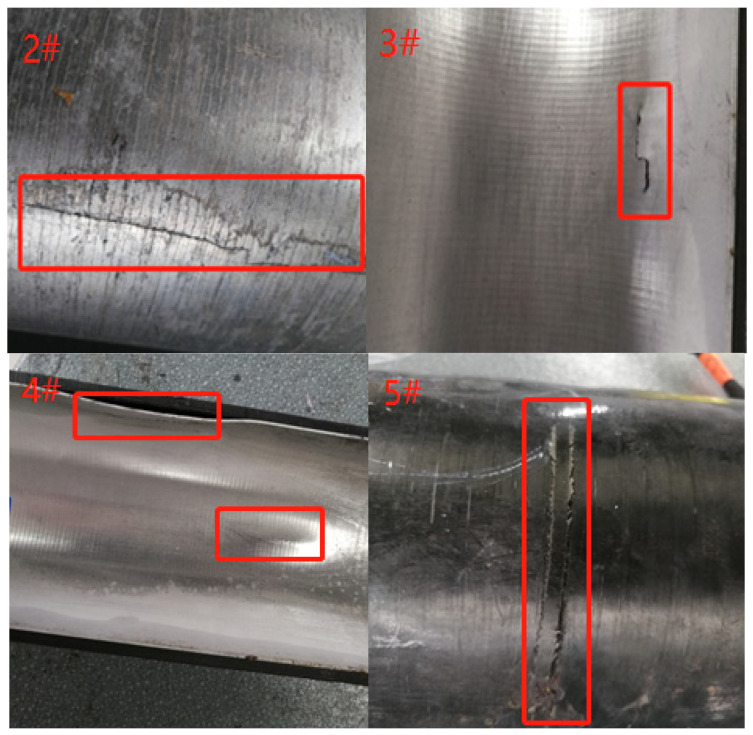
Fatigue failure location and mode of COPVs 2#–5#.

**Table 1 materials-16-02713-t001:** Material properties.

Project	E_1_ (GPa)	E_2_ (GPa)	G_12_ (GPa)	V_12_	V_23_	X_t_ (MPa)
6061 AL	70	70	26.9	0.3	0.3	262
T700SC/epoxy	154	114	7.09	0.33	0.49	2300
T700SC	230	-	-	-	-	4900

**Table 2 materials-16-02713-t002:** Design parameters and test results of COPVs 1# and 2#.

NO.	Total Number of Winding Layers	*K*	*n_a_*	*n_z_*	*λ*	Maximum Stress, MPa	Mean Burst Pressure, MPa
1#	42	0.75	28	14	2.0	2189.46	138
2#	48	0.65	32	16	2.0	1977.26	143

**Table 3 materials-16-02713-t003:** Design parameters and test results of COPVs 2#–5#.

NO.	Total Number of Winding Layers	*K*	*n_a_*	*n_z_*	*λ*	Maximum Stress, MPa	Mean Burst Pressure, MPa
2#	48	0.65	32	16	2.0	1977.26	143
3#	48	0.65	34	14	2.4	1716.77	155
4#	48	0.65	36	12	3.0	1614.55	170
5#	48	0.65	38	10	3.8	1935.90	158

**Table 4 materials-16-02713-t004:** Design parameters and test results of COPVs 2#–5#.

NO.	Total Number of Winding Layers	*λ*	Failure Location	Failure Mode	Fatigue Life Cycle
2#	48	2.0	cylindrical section	longitudinal crack	2852
3#	48	2.4	cylindrical section	longitudinal crack	5520
4#	48	3.0	cylindrical section	longitudinal crack	10,122
5#	48	3.8	cylindrical section	circular crack	7625

## Data Availability

The original data of this paper will not be shared externally because these specific data are integral to the next study.

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
