# Peer review of "Optimization of the Winding Layer Structure of High-Pressure Composite Overwrapped Pressure Vessels"

_materials, 2023, doi:10.3390/ma16072713_

Round 1

Reviewer 1 Report

The problem is a very common problem in industrial applications and a lot of similar research has been done. It may be more impressive if the difference in the study is explained in the Method Discussion section. The method used is an application that will eliminate the problem. Finite element analysis was carried out with appropriate methods. The data are valid and reliable. However, the study does not reveal a new method or technique. It is similar to the methods used in practice. Confirmation of the results obtained by the experimental application with finite element analysis may contribute to the literature. The graphics and tables presented in the study are at a sufficient level. However, the cited numbers and updatedness should be completely reviewed.

Introduction should be improved. More resources should be explored. The innovation revealed in this study should be compared with other studies. Finite element studies alone are not enough. Work can be strengthened by methods such as optimization or high computation. Parameters such as winding thickness, design pressure, production methods were used in the study. The content of these techniques needs further explanation.   The number of references should be increased. More emphasis should be placed on recent work. Support can be obtained from review articles on this subject.

Author Response

Dear reviewer,

Thank you for your suggestion. According to your suggestion, we have made some modifications, please review.

The problem is a very common problem in industrial applications and a lot of similar research has been done. It may be more impressive if the difference in the study is explained in the Method Discussion section. The method used is an application that will eliminate the problem. Finite element analysis was carried out with appropriate methods. The data are valid and reliable. However, the study does not reveal a new method or technique. It is similar to the methods used in practice. Confirmation of the results obtained by the experimental application with finite element analysis may contribute to the literature. The graphics and tables presented in the study are at a sufficient level. However, the cited numbers and updatedness should be completely reviewed.

Introduction should be improved. More resources should be explored. The innovation revealed in this study should be compared with other studies. Finite element studies alone are not enough. Work can be strengthened by methods such as optimization or high computation. Parameters such as winding thickness, design pressure, production methods were used in the study. The content of these techniques needs further explanation.   The number of references should be increased. More emphasis should be placed on recent work. Support can be obtained from review articles on this subject.

Re: The introduction and other contents have been revised. the content of “The design tools mainly include ANSYS、ABAQUS and other finite element analysis software for performance evaluation and optimization[7-9], which is also a very necessary method for the manufacture of COPV…” " Moreover, in many researches[21-23] on COPV, theoretical formulas and finite element software are used to design and predict the performance, and then the COPV performance is measured to verify the accuracy of theoretical design and simulation. Few studies can discuss from the perspective of the difference between theoretical design and practical performance, and give effective methods to combine design and practical experience……." are added.

I am very sorry that due to the word limit of the article, it is difficult to increase the content of the introduction.

See the revised article for details,please.

Reviewer 2 Report

The paper presents a rather interesting engineering issue. The work is written in an understandable form, and the content does not require significant corrections. However, the paper needs a number of corrections especially in terms of the quality of the presentation of results and in terms of editing. After addressing some of the comments below, the work can be published:

1. in the introduction, the novelty of the present work in relation to thematically similar other research works should be clearly presented.
2. the introduction is far too short and there is too little literature review. Please expand the introduction to include more literature items and it would be appropriate to additionally include papers in the field of research on composite materials (DOI: 10.1016/j.compstruct.2021.113597 and 10.1016/j.compstruct.2020.112716)
3. figure 1 is too small - please correct this.
4. Table 1 has editing shortcomings (please remove the spaces between the beginning of the brackets and the GPa information) for Young's modulus E1, E2 and G12.
5. formulas 1-7 have staggered relative to the text - please center them relative to the text.
6. Please be sure to enlarge Figure 3 and replace the indicators on the graph from "stars" to points.
7. The description of the content in section 2.5 is a joke. Please take the subject seriously and properly describe section 2.5 as befits a description of numerical models. First of all, please present the boundary conditions, the type of finite elements and more information. You can follow the previously mentioned works - which should be included in the introduction. This must be corrected.
8.You should never present results from FEM as Von Mises stresses.... especially for Laminates. Please study what type of stresses are shown for these types of composites.... The results in Figure 5 are inadequately presented - Mises stress is not shown for Laminates !
9. in Figure 7 also should not be Mises stresses... as already it is stresses on the main directions of orthotropy.... S11, S22 etc....
10. in Figure 9, the values on the graph will change, because all the figures from FEM simulations need to be corrected (they cannot represent Mises stresses !). Please make sure to update the changes in the figures from FEM in the text too. The work will be thoroughly reviewed after corrections.
11.The conclusions refer only to the qualitative assessment.... there is no reference to the Quantitative evaluation of the test results which is unacceptable.
12. the paper needs thorough revisions and will be reconsidered for publication after the revisions are made.

Author Response

Dear reviewer,

Thank you for your suggestion. According to your suggestion, we have made some modifications, please review.

  1. in the introduction, the novelty of the present work in relation to thematically similar other research works should be clearly presented.

Re: The introduction and other contents have been revised. the content of “The design tools mainly include ANSYS、ABAQUS and other finite element analysis software for performance evaluation and optimization[7-9], which is also a very necessary method for the manufacture of COPV…” " Moreover, in many researches[21-23] on COPV, theoretical formulas and finite element software are used to design and predict the performance, and then the COPV performance is measured to verify the accuracy of theoretical design and simulation. Few studies can discuss from the perspective of the difference between theoretical design and practical performance, and give effective methods to combine design and practical experience……." are added.

I am very sorry that due to the word limit of the article, it is difficult to increase the content of the introduction.See the revised article for details,please.

  1. the introduction is far too short and there is too little literature review. Please expand the introduction to include more literature items and it would be appropriate to additionally include papers in the field of research on composite materials (DOI: 10.1016/j.compstruct.2021.113597 and 10.1016/j.compstruct.2020.112716)

Re: Thank you very much for recommending the article, which has been quoted.

  1. figure 1 is too small - please correct this.

Re: Figure 1 has been enlarged.

  1. Table 1 has editing shortcomings (please remove the spaces between the beginning of the brackets and the GPa information) for Young's modulus E1, E2 and G12.

Re: It has been changed as required.

  1. formulas 1-7 have staggered relative to the text - please center them relative to the text.

Re: It has been changed as required.

  1. Please be sure to enlarge Figure 3 and replace the indicators on the graph from "stars" to points.

Re: It has been changed as required.

  1. The description of the content in section 2.5 is a joke. Please take the subject seriously and properly describe section 2.5 as befits a description of numerical models. First of all, please present the boundary conditions, the type of finite elements and more information. You can follow the previously mentioned works - which should be included in the introduction. This must be corrected.

Re: The following has been added to this section:

“In the finite element model, SOLID95 element and SHELL181 element are used to an-alyze the liner and wound layer respectively. Symmetry constraint is applied on the profile of the cylinder model, axial displacement constraint and rotation constraint of the other two axes are applied to the mouth of the cylinder……”

  1. You should never present results from FEM as Von Mises stresses.... especially for Laminates. Please study what type of stresses are shown for these types of composites.... The results in Figure 5 are inadequately presented - Mises stress is not shown for Laminates !

Re: Thank you very much. The Mises stress in the article has been changed to the first principal stress for analysis, and other parts have been changed accordingly.

  1. in Figure 7 also should not be Mises stresses... as already it is stresses on the main directions of orthotropy.... S11, S22 etc....
  2. in Figure 9, the values on the graph will change, because all the figures from FEM simulations need to be corrected (they cannot represent Mises stresses !). Please make sure to update the changes in the figures from FEM in the text too. The work will be thoroughly reviewed after corrections.

Re: Thank you. I have changed it.

11.The conclusions refer only to the qualitative assessment.... there is no reference to the Quantitative evaluation of the test results which is unacceptable.

Re: The conclusions have been modified as follows:

Conclusions

1、According to the netting theory formula, the design parameters of 70 MPa COPV were preliminarily determined, and the failure location of COPV was predicted by ANSYS finite element analysis method.

2、It is found that the measured performance of COPV is much different from the design goal, and the effect is little when the thickness of the winding layer is increased. By analyzing the failure mode, the ratio of annular fiber to helical fiber is further ad-justed, which greatly improves the bearing capacity and fatigue performance of COPV. When the number of winding layers is 48 and the ratio of annular fiber to helical fiber is 3.0, the performance of COPV is optimal. 

3、The method is proved to be reasonable by deducing the netting theory formula. At the same time, how to use the finite element simulation method to further design and predict the performance of COPV with large thickness needs to be further studied.

  1. the paper needs thorough revisions and will be reconsidered for publication after the revisions are made.

Reviewer 3 Report

In this paper, the fiber winding parameters of composite overwrapped pressure vessel
(COPV) is determined based on the netting theory, and the bearing capacity
. It was found that fatigue property of COPV is improved by adjusting the proportion of annular fiber and helical fiber. It also showed that for COPV with large thickness, the method of simply increasing the thickness of winding layer has poor effect on performance improvement. The paper has some interesting results that could make it publishable in the journal of Materials after the following major revisions:

1-The language of the paper in requires revision.

2-The first two sentences of the abstract are badly written. Especially this sentence needs to be rewritten.

In this paper, the fiber winding parameters of composite overwrapped pressure vessel (COPV) is determined based on the netting theory, and the bearing capacity and fatigue property of COPV is improved by adjusting the proportion of annular fiber and helical fiber, and the finite element analysis and performance testing are carried out to verify.”

3-Abstract needs to be modified. Start the abstract with what has been done theoretically and experimentally and then talk the main results.

4-Introduciton should be strengthened. Some old references are used. To modify this section the following documents can be consulted:

-(2022). Effects of Ni-decorated reduced graphene oxide nanosheets on the microstructural evolution and mechanical properties of Sn-3.0Ag-0.5Cu composite solders. Intermetallics, 150, 107683. doi: https://doi.org/10.1016/j.intermet.2022.107683

-(2020). Preparation of PI porous fiber membrane for recovering oil-paper insulation structure. Journal of materials science. Materials in electronics, 31(16), 13344-13351. doi: 10.1007/s10854-020-03888-5

-(2017). Ameliorated longitudinal critically refracted—Attenuation velocity method for welding residual stress measurement. Journal of Materials Processing Technology, 246, 267-275. doi: https://doi.org/10.1016/j.jmatprotec.2017.03.022

5-better define the novelty of the work at the end of introduction.

6-figure 1 could be deleted.

7-instead of materials properties of table 1, write their chemical compositions.

8-Consult the following document in the discussion section.

-(2022). A Ferrotoroidic Candidate with Well‐Separated Spin Chains. Advanced materials (Weinheim), 34(12), e2106728. doi: 10.1002/adma.202106728

-(2022). High-Performance Neuromorphic Computing Based on Ferroelectric Synapses with Excellent Conductance Linearity and Symmetry. Advanced Functional Materials, 32(35), 2202366. doi: https://doi.org/10.1002/adfm.202202366

9-figure 4 doesn’t mean anything. It should be deleted.

10-put error bars for figure 9.

11-figure 10 should be deleted. It is only a thickness.

12-It is “conclusions” not conclusion and put them as bullet points.

Author Response

Dear reviewer,

Thank you for your suggestion. According to your suggestion, we have made some modifications, please review.

  • The language of the paper in requires revision.

Re: It has been modified and adjusted. For details, see the revised article.

2-The first two sentences of the abstract are badly written. Especially this sentence needs to be rewritten.

“In this paper, the fiber winding parameters of composite overwrapped pressure vessel (COPV) is determined based on the netting theory, and the bearing capacity and fatigue property of COPV is improved by adjusting the proportion of annular fiber and helical fiber, and the finite element analysis and performance testing are carried out to verify.”

  • Abstract needs to be modified. Start the abstract with what has been done theoretically and experimentally and then talk the main results.

Re: The abstract has been changed to: “The large thickness COPV is designed by netting theory and finite element simulation method, but the actual performance is low, and the cylinder performance still cannot be improved after increasing the thickness of the composite winding layer. This paper analyzes the reasons and puts forward a feasible method: without changing the thickness of winding layer, the performance of COPV can be effectively in-creased by increasing the proportion of annular winding fiber. The method is verified by test and supported by theory.”

  • Introduciton should be strengthened. Some old references are used. To modify this section the following documents can be consulted:

Re: The introduction has been modified and enhanced, the content of “The design tools mainly include ANSYS、ABAQUS and other finite element analysis software for performance evaluation and optimization[7-9], which is also a very neces-sary method for the manufacture of COPV…” " Moreover ,in many researches[21-23] on COPV, theoretical formulas and finite element software are used to design and predict the performance, and then the COPV perfor-mance is measured to verify the accuracy of theoretical design and simulation. Few studies can discuss from the perspective of the difference between theoretical design and practical performance, and give effective methods to combine design and practical experience……." are added.

See the revised article for details,please.

Some of the literature recommended below has been added as citations.

-(2022). Effects of Ni-decorated reduced graphene oxide nanosheets on the microstructural evolution and mechanical properties of Sn-3.0Ag-0.5Cu composite solders. Intermetallics, 150, 107683. doi: https://doi.org/10.1016/j.intermet.2022.107683

-(2020). Preparation of PI porous fiber membrane for recovering oil-paper insulation structure. Journal of materials science. Materials in electronics, 31(16), 13344-13351. doi: 10.1007/s10854-020-03888-5

-(2017). Ameliorated longitudinal critically refracted—Attenuation velocity method for welding residual stress measurement. Journal of Materials Processing Technology, 246, 267-275. doi: https://doi.org/10.1016/j.jmatprotec.2017.03.022

  • better define the novelty of the work at the end of introduction.

Re: This section is adjusted to “……few studies can discuss from the perspective of the difference between theoretical de-sign and practical performance, and give effective methods to combine design and practical experience.

In this work, by comparing the theoretical calculation, finite element simulation results and actual test performance of COPV, based on the failure mode of COPV, a method to optimize the performance of large thickness COPV is proposed, which pro-vides design ideas and experience parameters for the design and manufacture of 70MPa COPV."

  • figure 1 could be deleted.

Re:Figure 1 has been deleted.

7-instead of materials properties of table 1, write their chemical compositions.

Re:I am very sorry that the parameters in Table 1 are basic parameters of finite element analysis and cannot be replaced by other parameters.

8-Consult the following document in the discussion section.

Re:I am very sorry that these two literatures are not easy to be added because they are quite different from the research scope of this paper.

-(2022). A Ferrotoroidic Candidate with Well‐Separated Spin Chains. Advanced materials (Weinheim), 34(12), e2106728. doi: 10.1002/adma.202106728

-(2022). High-Performance Neuromorphic Computing Based on Ferroelectric Synapses with Excellent Conductance Linearity and Symmetry. Advanced Functional Materials, 32(35), 2202366. doi: https://doi.org/10.1002/adfm.202202366

9-figure 4 doesn’t mean anything. It should be deleted.

Re:Figure 4 has been deleted.

10-put error bars for figure 9.

Re:The item has been added to the figure.

11-figure 10 should be deleted. It is only a thickness.

Re: Figure 10 has been deleted.

12-It is “conclusions” not conclusion and put them as bullet points.

Re: Changed as requested.

The conclusions have been modified as follows:

Conclusions

1、According to the netting theory formula, the design parameters of 70 MPa COPV were preliminarily determined, and the failure location of COPV was predicted by ANSYS finite element analysis method.

2、It is found that the measured performance of COPV is much different from the design goal, and the effect is little when the thickness of the winding layer is increased. By analyzing the failure mode, the ratio of annular fiber to helical fiber is further ad-justed, which greatly improves the bearing capacity and fatigue performance of COPV. When the number of winding layers is 48 and the ratio of annular fiber to helical fiber is 3.0, the performance of COPV is optimal. 

Round 2

Reviewer 1 Report

Acceted

Reviewer 2 Report

The paper can be published.

Reviewer 3 Report

The paper can be published in its revised format.